# Discovery and biological evaluation of a potent small molecule CRM1 inhibitor for its selective ablation of extranodal NK/T cell lymphoma

He Liu[1†], Meisuo Liu[1†], Xibao Tian[1†], Haina Wang[1†], Jiujiao Gao[1†], Hanrui Li[1], Zhehuan Zhao[2], Yu Liu[2], Caigang Liu[3*], Xuan Chen[1,2], Yongliang Yang[1*]

[1]School of Bioengineering, Dalian University of Technology, Dalian, China; [2]School of Software, Dalian University of Technology, Dalian, China; [3]Department of Oncology, Shengjing Hospital of China Medical University, Shenyang, China

## Abstract

**Background:** The overactivation of NF-κB signaling is a key hallmark for the pathogenesis of extranodal natural killer/T cell lymphoma (ENKTL), a very aggressive subtype of non-Hodgkin's lymphoma yet with rather limited control strategies. Previously, we found that the dysregulated exportin-1 (also known as CRM1) is mainly responsible for tumor cells to evade apoptosis and promote tumor-associated pathways such as NF-κB signaling.

**Methods:** Herein we reported the discovery and biological evaluation of a potent small molecule CRM1 inhibitor, LFS-1107. We validated that CRM1 is a major cellular target of LFS-1107 by biolayer interferometry assay (BLI) and the knockdown of CRM1 conferred tumor cells with resistance to LFS-1107.

**Results:** We found that LFS-1107 can strongly suppresses the growth of ENKTL cells at low-range nanomolar concentration yet with minimal effects on human platelets and healthy peripheral blood mononuclear cells. Treatment of ENKTL cells with LFS-1107 resulted in the nuclear retention of IκB$_\alpha$ and consequent strong suppression of NF-κB transcriptional activities, NF-κB target genes downregulation and attenuated tumor cell growth and proliferation. Furthermore, LFS-1107 exhibited potent activities when administered to immunodeficient mice engrafted with human ENKTL cells.

**Conclusions:** Therefore, LFS-1107 holds great promise for the treatment of ENKTL and may warrant translation for use in clinical trials.

**Funding:** Yang's laboratory was supported by the National Natural Science Foundation of China (Grant: 81874301), the Fundamental Research Funds for Central University (Grant: DUT22YG122) and the Key Research project of 'be Recruited and be in Command' in Liaoning Province (Personal Target Discovery for Metabolic Diseases).

**\*For correspondence:**
angel-s205@163.com (CL);
everbright99@foxmail.com (YY)

†These authors contributed equally to this work

## Editor's evaluation

This study provides important findings on the discovery of a novel CRM1 inhibitor. This study provides compelling data on the identification of a novel CRM1 inhibitor and shows its efficiency against extranodal natural killer/T cell lymphoma cells (ENKTL). The findings from this study will provide motivation on optimizing these novel CRM inhibitors for treatments of ENKTL and will be of broad interest to cancer biology.

## Introduction

ENKTL represents as a rare yet highly aggressive subtype of non-Hodgkin's lymphoma (NHL) (*Yamaguchi et al., 2018*) with rather poor prognosis and short median survival time (*Suzuki et al., 2010*). Previously, we and colleagues uncovered that the overactivation of NF-κB signaling and its downstream cytokines is a key hallmark for the pathogenesis of ENKTL (*Wen et al., 2018*). Nevertheless, to date, targeted therapy towards ENKTL is still lacking and new therapeutic strategies are urgently needed to combat this rare yet deadly tumor (*Jiang et al., 2015*). Exportin-1, also known as CRM1, has been implicated in the aggressive behavior of various malignances by nuclear exporting critical tumor suppressor proteins and transcription factors (*Dong et al., 2009*). Moreover, CRM1 has been found to be highly upregulated in a panel of tumor types (*Turner et al., 2012*). Indeed, it is believed that tumor cells strive to evade powerful negative regulation of apoptosis and cell proliferation through CRM1-mediated nuclear export machinery. Previously, we reported that a natural and covalent CRM1 inhibitor sulforaphene and its synthetic analogues (*Wang et al., 2018*; *Tian et al., 2020*; *Liu et al., 2023*; *Gao et al., 2021*), can selectively kill a panel of tumor cells. Here, we presented the development and evaluation of a novel sulforaphene analogue as a potent CRM1 inhibitor with superior antitumor activities towards ENKTL while sparing human platelets.

## Results and discussion

In this work, we want to find aromatic fragments that can be installed with sulforaphene parent structure as CRM1 inhibitors with strong potency. Previously, we developed a CRM1 inhibitor LFS-829 via traditional virtual screening and structure-based design approach. Herein, we strive to design a CRM1 inhibitor with strong potency against ENKTL cells through the recently developed AIDD (Artificial Intelligence Drug Discovery) approaches (*Chan et al., 2019*). Briefly, we adopted the deep reinforcement learning molecular de novo design model developed by *Olivecrona et al., 2017* to facilitate the discovery of novel aromatic fragments. This model prioritizes those structures with modest similarity to the molecules in the positive dataset of known CRM1 inhibitors rather than very close analogues. We obtained an initial output of 3,000 moiety structures and the top 50 candidate moieties generated from this step were used as seed structures to search for commercial-available aromatic fragments from Sigma-Aldrich compound library (similarity ~97%). Next, we selected and purchased 10 commercial-accessible aromatic fragments (*Figure 1A*) which were experimentally tested via biolayer interferometry assay (BLI). Among the ten experimentally tested fragments, tetrazole moieties (S5 and S8 fragment) obtained the second highest binding constant with CRM1 via BLI assay experiments (*Figure 1C* and *Figure 1—figure supplement 1*). Yet, given that tetrazole fragments are more synthetic accessible as compared to the fragment with the highest binding constant (S4 moiety in *Figure 1A*, benzo[d]oxazol-2-yl)-N, N-dimethylaniline), the tetrazole moiety was chosen to install with sulforaphene parent structure. Subsequently, we prepared a synthetic analogue of sulforaphene with the tetrazole aromatic moiety, named as LFS-1107 (*Figure 1B*), which was further evaluated through cell lines and in vivo animal models.

First, we sought to evaluate the binding specificity of LFS-1107 with CRM1 as compared to other possible protein targets with reactive cysteine in the binding pocket through the Octet K2 biolayer interferometry assay (BLI). Here, we chose IkB$_\alpha$ and Keap1 as two control probes to compare because both of them are reactive for covalent compounds and thereby frequently used for testing selectivity. Our BLI results revealed that LFS-1107 binds strongly with CRM1 ($K_d$~1.25E-11 M, *Figure 1D*) as compared to KPT-330 ($K_d$~5.29E-09 M), a known CRM1 inhibitor approved by FDA. In contrast, LFS-1107 doesn't bind with either two control probes IkB$_\alpha$ or Keap1 in BLI assay experiments (*Figure 1—figure supplement 2*). Furthermore, the results revealed that LFS-1107 is a reversible CRM1 inhibitor with clear dissociation process which may implicate a low toxicity profile. This is consistent with the results of our previous studies that sulforaphene synthetic analogues are reversible CRM1 inhibitors. Moreover, we used GSH/GSSG detection assay kit to show that the GSH:GSSG ratio keeps constant upon the treatment of LFS-1107 and therefore we ruled out the possibility that LFS-1107 is a general redox modulator (*Figure 2B*). Collectively, our results revealed that CRM1 is a major cellular target of LFS-1107 and responsible for its cellular activities.

Next, we assessed LFS-1107 for its activity and specificity in human ENKTL cell lines. Our data show that LFS-1107 achieves IC$_{50}$ value of 26 nM in SNK6 cell line and 36 nM in HANK-1 cell line (*Figure 2A*).

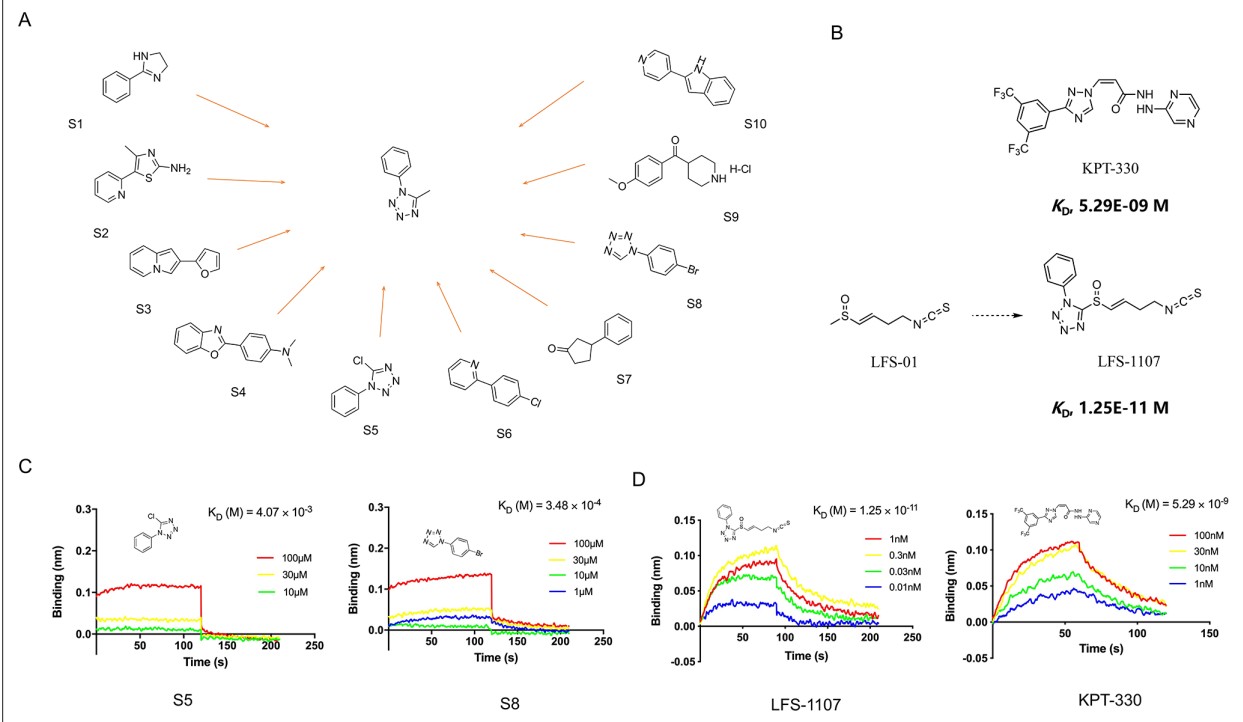

**Figure 1.** The discovery of sulforaphene synthetic analogue LFS-1107. (**A**) The identification of ten commercial-accessible aromatic fragments aided by deep reinforcement learning model; (**B**) Synthesis of LFS-1107 via the installation of aromatic tetrazole moiety selected from the previous step to the sulforaphene parent structure; (**C**) Assessment of protein-ligand binding kinetics and binding affinity of tetrazole aromatic fragments via Bio-layer interferometry (BLI) assay; (**D**) Binding affinity of LFS-1107 and KPT-330 determined via BLI assay: LFS-1107, $K_d$~1.25E-11 M; KPT-330: $K_d$~5.29E-09 M.

The online version of this article includes the following source data and figure supplement(s) for figure 1:

**Source data 1.** The chemical structure of 10 commercial-accessible aromatic fragments.

**Source data 2.** The synthesis of compound LFS-1107.

**Source data 3.** The data of affinities and binding kinetics of CRM1 to S5 and S8.

**Source data 4.** The data of affinities and binding kinetics of CRM1 to LFS-1107 and KPT-330.

**Figure supplement 1.** Binding affinities and binding kinetics of ten commercial-accessible fragments with CRM1 were determined using Bio-layer interferometry (BLI) assay.

**Figure supplement 1—source data 1.** The data of affinities and binding kinetics of CRM1 to ten commercial-accessible aromatic fragments.

**Figure supplement 2.** The BLI results of two control proteins Keap1 (**A**) and IκBα (**B**).

**Figure supplement 2—source data 1.** The data of affinities and binding kinetics of Keap1 and IκBα to LFS-1107.

**Figure supplement 3.** Organic synthesis scheme of compound LFS-1107.

**Figure supplement 3—source data 1.** Organic synthesis scheme of compound LFS-1107.

**Figure supplement 4.** Expression of CRM1 mRNA in different tumor types.

**Figure supplement 4—source data 1.** The expression of CRM1 mRNA in different tumor types.

To further study the selectivity of LFS-1107 towards ENKTL cells, we evaluated the toxicity of LFS-1107 in normal PBMCs isolated from peripheral blood of healthy donors (*Figure 2C*). Interestingly, LFS-1107 barely showed any toxicity at concentration of 4 μM and the toxicity towards normal human PBMC is minimal even at a high dose of 9 μM. This implicates that LFS-1107 can selectively eliminate ENKTL cells while sparing normal human PBMC with good safety profile. Moreover, drug-induced toxicity towards platelets (also called as thrombocytopenia) is a common side effect in clinical trials for cancer therapeutics. We assessed the toxicity effects of LFS-1107 towards platelets and we found that LFS-1107 barely exhibits any effects on human platelets even at very high concentrations of 500 μM (*Figure 2D*).

Subsequently, we want to investigate whether LFS-1107 could inhibit CRM1-mediated nuclear export of IκBα and reduce constitutive NF-κB activity in ENKTL cells. First, we demonstrated

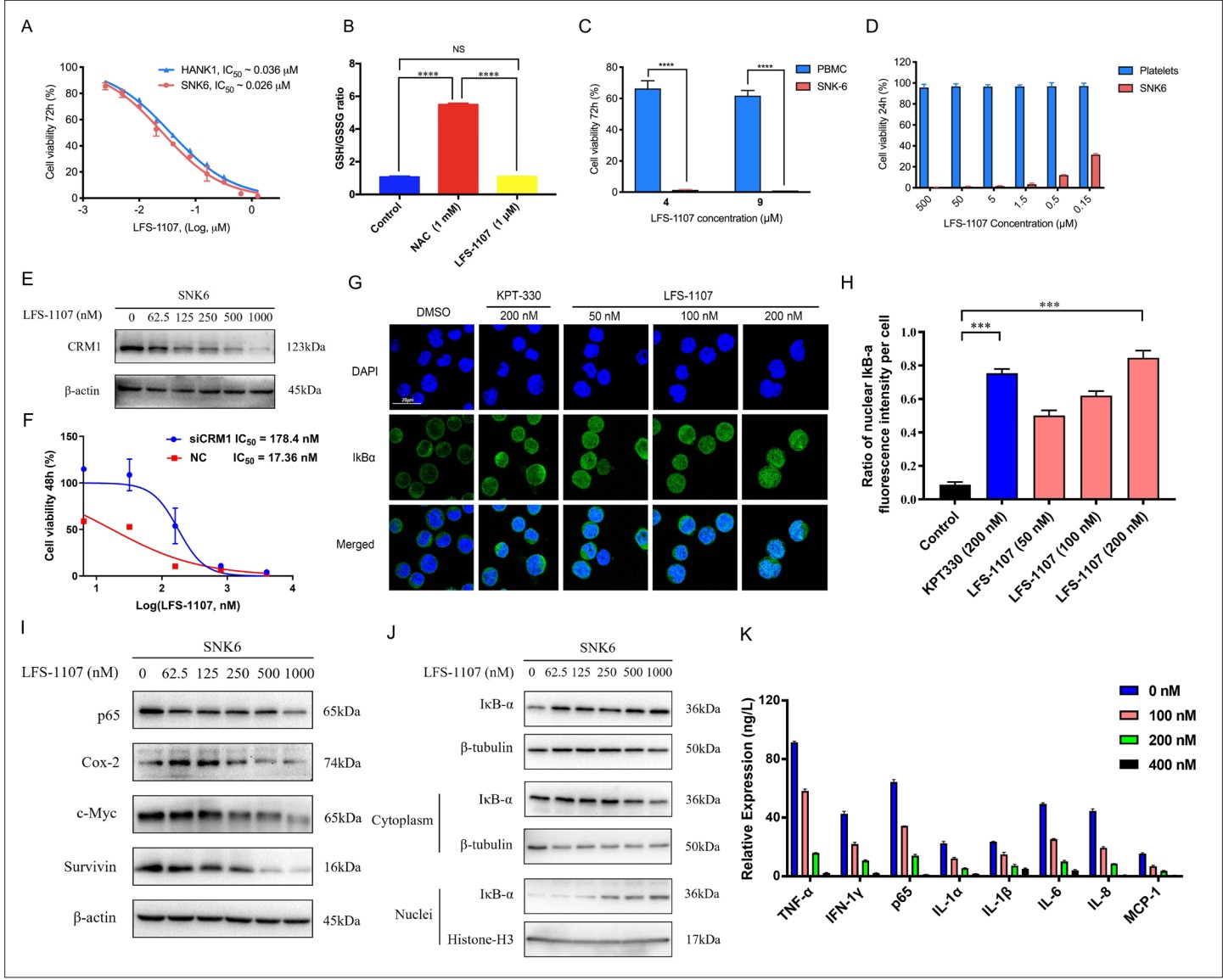

**Figure 2.** LFS-1107 strongly suppresses the growth of ENKTL cells acting through the nuclear retention of IkB$_\alpha$ and subsequent attenuation of NF-$\kappa$B signaling. (**A**) Suppression of different human NK/T cell lymphoma cells; (**B**) GSH/GSSG ratio detection upon the treatment of LFS-1107; (**C**) The effect of LFS-1107 on normal PBMC cell lines; (**D**) The viability of platelets treated with LFS-1107; (**E**) Western blot result of the CRM1 with β-actin as loading control; (**F**) The cellular activities of LFS-1107 on siCRM1-293T and the wild-type 293T cell line with different concentrations of treatment for 48 h; (**G**) Nuclear retention of I$\kappa$B$\alpha$ by confocal microscopy (Scale bar, 20 μM); (**H**) Quantification of the nuclear I$\mathrm{K}$B-$\alpha$ ratio by fluorescence intensity per cell (Data presented as mean ± SEM); (**I**) Representative western blots of p65, Cox-2, c-Myc, and Survivin; (**J**) Western blots showing the protein level of I$\kappa$B$\alpha$ in nucleus and cytoplasm; (**K**) ELISA detection of cytokine production after treated with LFS-1107. Data were either presented as representative images or expressed as the mean ± SD of each group. Statistical analysis were performed via Student t-test, ***p<0.001, ****p<0.0001.

The online version of this article includes the following source data and figure supplement(s) for figure 2:

**Source data 1.** Inhibition of the cell growth of SNK6 and Hank-1 cells by LFS-1107.

**Source data 2.** GSH/GSSG ratio detection upon the treatment of LFS-1107.

**Source data 3.** Suppression of the cell growth of PBMC cells by LFS-1107.

**Source data 4.** Suppression of the cell growth of platelets by LFS-1107.

**Source data 5.** Immunoblot of CRM1 expression after LFS-1107 treatment.

**Source data 6.** The cellular activities of LFS-1107 on siCRM1-293T and the wild-type 293T cell line.

**Source data 7.** Nuclear accumulation of I$\kappa$B$\alpha$ induced by treatment with LFS-1107 for 3 hr.

**Source data 8.** Quantification of the nuclear I$\kappa$B$\alpha$ ratio by fluorescence intensity per cell.

*Figure 2 continued on next page*

*Figure 2 continued*

**Source data 9.** Immunoblot of expression p65, Cox-2, c-Myc, and Survivin after LFS-1107 treatment.

**Source data 10.** Immunoblot of I κ Bα in nucleus and cytoplasm expression after LFS-1107.

**Source data 11.** ELISA detection of TNF-α, IFN-1γ, p65, IL-1α, IL-1β, IL-6, IL-8, and MCP-1 after treated with LFS-1107.

**Figure supplement 1.** Proteomics analysis of SNK6 cells (control vs. LFS-1107 treatment) indicates that CRM1 was downregulated upon LFS-1107 treatment (FC~1.3).

**Figure supplement 1—source data 1.** Proteomics analysis of SNK6 cells.

**Figure supplement 2.** Cellular activities of KPT-330 towards two ENKTL cell lines.

**Figure supplement 2—source data 1.** Cellular activities of KPT-330 towards two ENKTL cell lines.

**Figure supplement 3.** Representative immunofluorescence images of I κ Bα (stained with Cy3) localization in Hela cells.

**Figure supplement 3—source data 1.** Nuclear accumulation of I κ Bα induced by treatment with 500 nM LFS-1107 for 3 hr.

that LFS-1107 could suppress the expression of CRM1 in a dose-dependent manner (*Figure 2E*). We revealed that the cellular activities by LFS-1107 was significantly abrogated when CRM1 was knocked down by siCRM1, implicating that CRM1 is a main target responsible for the cellular activities of LFS-1107 (*Figure 2F*). Our immunofluorescent results by confocal microscopy revealed that LFS-1107 can lead to nuclear accumulation of IκBα after 3 hr treatment in a dose dependent manner (*Figure 2G–H*). Next, we employed Western blot to further confirm nuclear localization of IκBα upon the treatment of LFS-1107 (*Figure 2J*). The results ascertained that IκBα was trapped in the nucleus as the protein level of nuclear IκBα was significantly upregulated after LFS-1107 treatment (*Figure 2J*). Moreover, it is well known that the constitutive activity of NF-κB/p65 was accompanied by the increased production of a few proinflammatory cytokines. Consequently, we found the expression of proinflammatory and proliferative proteins p65, COX-2, c-Myc, and Survivin were downregulated in a dose dependent manner after treatment with LFS-1107 (*Figure 2I*). Furthermore, when SNK6 cells were treated with LFS-1107, we observed a significant reduction of the proinflammatory cytokines including TNF-α, IFN-γ, NF-κB/p65, IL-1α, IL-1β, IL-6, IL-8, and MCP-1 as measured by ELISA assay (*Figure 2K*). Moreover, proteomics analysis suggests that Biological Process (regulation of cellular component organization) and Molecular Function (protein binding) related to CRM1 were modulated upon the treatment of LFS-1107 (*Figure 2—figure supplement 1*). These results suggested that the inhibition of CRM1 by LFS-1107 could lead to the downregulation of NF-κB transcriptional activity, proinflammatory cytokines and oncogenic signatures of ENKTL.

Lastly, we established a xenograft mouse model to test our findings by injecting SNK6 cells intraperitoneally into NCG mice. We found that LFS-1107 treatment (10 mg/kg/week) was able to extend mouse survival (*Figure 3A–B*) and eliminate tumor cells considerably as evidenced by flow cytometric analysis (*Figure 3C–D–E*), demonstrating the efficacy of LFS-1107 in controlling ENKTL. Moreover, remarkably, mice injected with LFS-1107 reduced splenomegaly and restored spleen weight and spleen volume as compared with the normal group (*Figure 3F–G*) with a better overall survival rate. Noteworthy, splenomegaly is a conventional symptom for extranodal natural NK/T cell lymphoma patients (*Tse and Kwong, 2017*). Hence, our results implicate that LFS-1107 can ameliorate the symptoms of ENKTL and might be a potentially promising compound for ENKTL treatment.

## Conclusions

In summary, we report the discovery and biological evaluation of a synthetic sulforaphene analogue as potent CRM1 inhibitor towards the treatment of ENKTL. We demonstrated that treatment of LFS-1107 holds great promise as an effective remedy for ENKTL, acting through the nuclear retention of IκBα and subsequent attenuation of NF-κB signaling. We want to remind the reader that we don't rule out the possibility that LFS-1107 may kill ENKTL cells via other mechanisms due to the broad-spectrum cargo proteins of CRM1. Yet, we want to argue that our study has captured the most prominent mechanism of actions for LFS-1107 towards ENKTL. Our study may represent a novel route for eliminating ENKTL tumor cells with a distinct mode of actions.

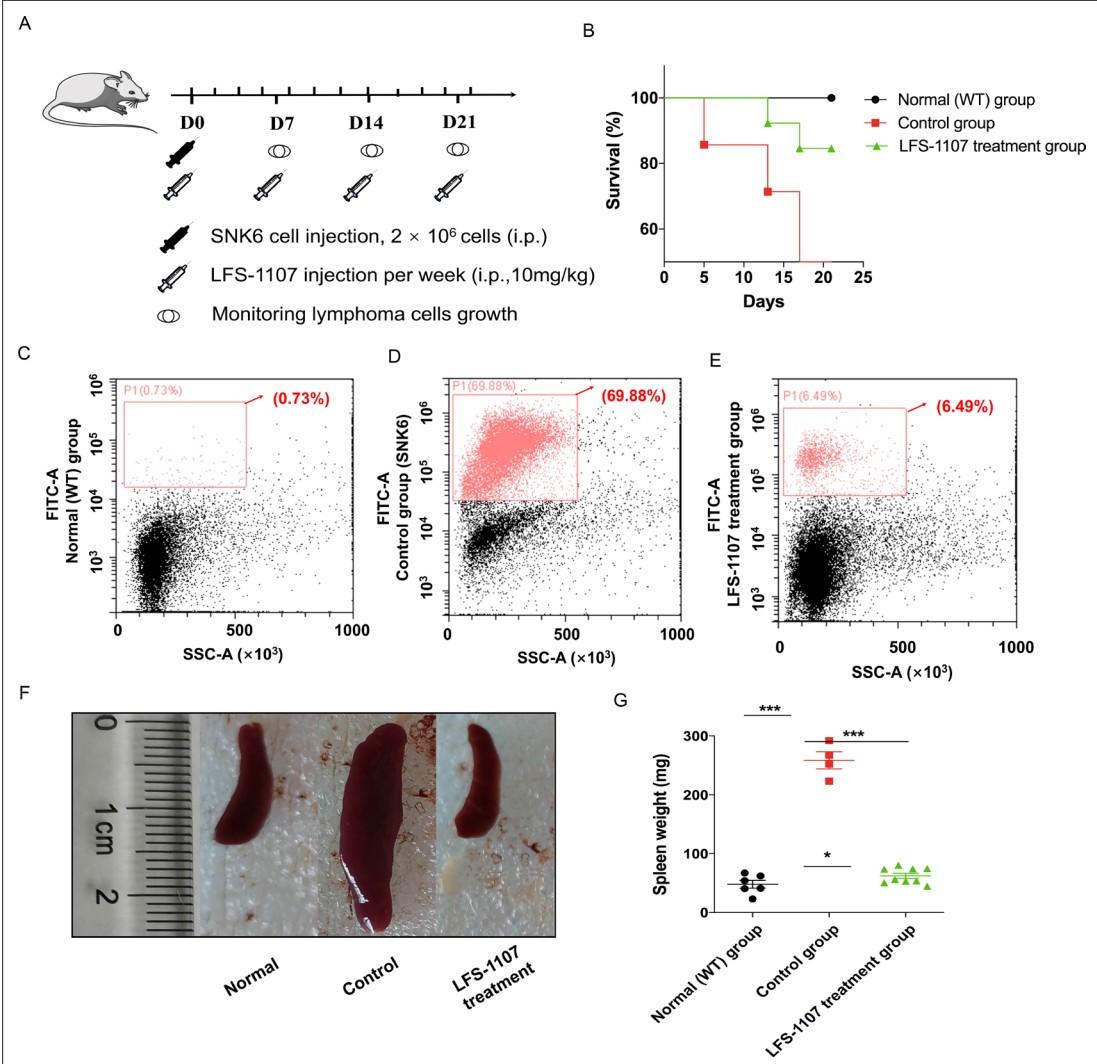

**Figure 3.** LFS-1107 can ameliorate the symptoms of ENKTL in xenograft mouse model. (**A**) Scheme of the xenograft mouse model. The mice were randomly divided into three groups (n = 6-13 per group). SNK6 cells were injected i.v. in female NOD SCID mice, and mice were injected per week with LFS-1107; (**B**) Survival rate of different animal groups; (**C–E**) Flow cytometry of human ENKTL cell lines in mouse bone marrow with the use of FITC anti-human CD45 antibody (P1, red) or no primary antibody control (black); (**F–G**) The symptoms of splenomegaly in each group of euthanized mice. From left to right: normal group, control ENKTL xenograft model group(splenomegaly), LFS-1107 treatment group (10 mg/kg). Data were either presented as representative images or expressed as the mean ± SEM of each group. Statistical analysis were performed via Student t-test, *p<0.05, ***p<0.001.

The online version of this article includes the following source data for figure 3:

**Source data 1.** The process for the in the xenograft mouse model study.

**Source data 2.** Survival rate of normal group, control group and LFS-1107 treatment group.

**Source data 3.** Flow cytometry of human ENKTL cell lines in mouse bone marrow.

**Source data 4.** The symptoms of splenomegaly in normal group, control group and LFS-1107 treatment group mice.

**Source data 5.** The spleen weight of normal group, control group and LFS-1107 treatment group mice.

## Materials and methods

| Reagent type (species) or resource | Designation | Source or refrerence | Identifiers | Additional information |
|---|---|---|---|---|
| Cell line (*Homo sapiens*) | SNK-6 | ATCC | RRID:CVCL_A673 | |
| Cell line (*Homo sapiens*) | HANK-1 | ATCC | RRID:CVCL_8226 | |
| Cell line (*Homo sapiens*) | HEK293T | ATCC | RRID:CVCL_0063 | |
| Cell line (*Homo sapiens*) | Hela | ATCC | RRID:CVCL_0030 | |
| Antibody | Cox2 Antibody | Cell Signaling Technology, Danvers, MA | Cat# 4842 RRID:AB_2084968 | WB (1:1000) |
| Antibody | NF-κB p65 (D14E12) XP Rabbit mAb | Cell Signaling Technology, Danvers, MA | Cat# 8424 | WB (1:1000) |
| Antibody | c-Myc/N-Myc (D3N8F) Rabbit mAb | Cell Signaling Technology, Danvers, MA | Cat# 13987 RRID:AB_2631168 | WB (1:1000) |
| Antibody | Anti-Survivin antibody | Abcam, Waltham, MA | Cat# Ab76424 RRID:AB_1524459 | WB (1:1000) |
| Antibody | IκBα (L35A5) Mouse mAb (Amino-terminal Antigen) | Cell Signaling Technology, Danvers, MA | Cat# 4814 RRID:AB_390781 | WB (1:1000) IF(1:200) |
| Antibody | Exportin-1/CRM1 (D6V7N) Rabbit mAb | Cell Signaling Technology, Danvers, MA | Cat# 46249 RRID:AB_2799298 | WB (1:1000) |
| Antibody | Histone H3 (D1H2) XP Rabbit mAb (BSA and Azide Free) | Cell Signaling Technology, Danvers, MA | Cat# 60932 | WB (1:1000) |
| Antibody | CD45 (Intracellular Domain) (D9M8I) XP Rabbit mAb | Cell Signaling Technology, Danvers, MA | Cat# 13917 RRID:AB_2750898 | F (1:100) |
| Antibody | FITC Goat Anti-Mouse IgG (H+L) | ABclonal, Wuhan, China | Cat# AS001 RRID:AB_1524459 | IF(1:50) |
| Protein | KEAP1 Protein, Human, Recombinant (His & GST Tag) | Sinobiological, Beijing, China | Cat# 11981-H20B | |
| Protein | IκB alpha Protein, Human, Recombinant (His Tag) | Sinobiological, Beijing, China | Cat# 12045-H07E | |
| Commercial assay or kit | Cell Counting Kit-8 | Beyotime, Shanghai, China | Cat# C0037 | |
| Commercial assay or kit | GSH and GSSG Assay Kit | Beyotime, Shanghai, China | Cat# S0053 | |
| Commercial assay or kit | TNF-α Elisa kit | Elabscience, Wuhan, China | Cat# E-EL-H2305c | |
| Commercial assay or kit | IFN-1γ Elisa kit | Elabscience, Wuhan, China | Cat# E-EL-H0108c | |
| Commercial assay or kit | p65 Elisa kit | Elabscience, Wuhan, China | Cat# E-EL-H1388c | |
| Commercial assay or kit | IL-1α Elisa kit | Elabscience, Wuhan, China | Cat# E-EL-H0088c | |
| Commercial assay or kit | IL-1β Elisa kit | Elabscience, Wuhan, China | Cat# E-EL-H0149c | |
| Commercial assay or kit | IL-6 Elisa kit | Elabscience, Wuhan, China | Cat# E-EL-H6156 | |
| Commercial assay or kit | IL-8 Elisa kit | Elabscience, Wuhan, China | Cat# E-EL-H6008 | |
| Commercial assay or kit | MCP-1 Elisa kit | Elabscience, Wuhan, China | Cat# E-EL-H6005 | |
| Commercial assay or kit | NE-PER Nuclear and Cytoplasmic Extraction Reagents | Thermo, USA | Cat# 78833 | |
| Software, algorithm | GraphPad Prism | GraphPad Software | RRID:SCR_002798 | |
| Chemical compound, drug | KPT330 | Targetmol, USA | Cat# T1844 | CAS1421923-86-5 |
| Other | DAPI | Beyotime, Shanghai, China | Cat# C1005 | |

### The purity of the compound

Compound LFS-1107 was ascertained by HNMR, LC-MS and HPLC analysis (≥95% purity). The organic synthesis was conducted following standard procedures. All the chemicals and solvents were purchased from Sigma.

### Organic synthesis

The NMR spectra of small molecule compounds were recorded on Bruker Avance 400 MHz for 1HNMR and 100 MHz for 13CNMR. The LCMS were taken on a quadrupole Mass Spectrometer on Shimadzu

LCMS 2010 (Column: sepax ODS 50×2.0 mm, 5 μm) or Agilent 1200 HPLC, 1956 MSD (Column: Shim-pack XR-ODS 30×3.0, 2.2 μm) operating in ES (+) ionization mode. Chromatographic purifications were by flash chromatography using 100~200 mesh silica gel. Anhydrous solvents were pre-treated with 3 A MS column before used. All commercially available reagents were used as received unless otherwise stated.

## Cell lines, cell cultures, and antibodies

Human lymphoma cell lines SNK6 and HANK-1 were purchased from ATCC. SNK6 and HANK-1 cell lines were maintained in RPMI 1640 medium supplemented with 10% FBS and 293T cell line in DMEM medium with 10% FBS. RPMI 1640 medium, DMEM medium and FBS (fetal bovine serum) were purchased from Hyclone (Thermo Scientific). Antibodies against Survivin (ab76424) was obtained from Abcam. Other antibodies were purchased from Cell Signaling Technology: anti-β-actin (#3700), anti-Cox2 (#4842), anti-p65 (#8424), anti-C-Myc (#13987), and anti-IκBα (#4814). Alexa 488-conjugated goat anti-rabbit antibody and Alexa 594-conjugated goat anti-mouse antibody were obtained from Invitrogen Life Technology (Invitrogen, CA, USA). Human lymphoma cell lines SNK6, HANK-1, 293T and Hela were purchased from the American Type Culture Collection (ATCC). The identity of these cell lines was confirmed by STR and the cells were tested negative for mycoplasma contamination throughout the experimental period.

## Deep reinforcement learning model

In the present study, we adopted the deep reinforcement learning based molecular de novo design method developed by Olivecrona etc. from AstraZeneca. In brief, the deep reinforcement learning model consists of two main modules, the Design module (D) and the Evaluate module (E). The Design module (D) is used to produce novel chemical structures whereas the Evaluate module (E) is used to assess the feasibility and properties of the novel structures by assigning a numerical award or penalty to each new structure. The model employs the simple representation of chemical structures by the simplified molecular-input line-entry system (SMILES) strings in both the Design module (D) and the Evaluate module (E). The Design module was first pre-trained with about ~2 million structures from the PubChem database to learn basic rules of organic chemistry that define SMILES strings within the context of Recurrent neural network (RNN). To check the validity of the approach, we used the module to generate about 0.2 million compounds which were assessed by the structure checker ChemAxon and 97% of the generated structures were chemical sensible structures. Finally, the deep reinforcement learning based molecular design model was able to design ~3000 chemical structures from which we chose the top 50 candidate moieties for further consideration. In this work, we want to find novel fragments that can be installed with sulforaphene parent structure as CRM1 inhibitors. Indeed, our model prioritizes those structures with modest similarity to the molecules in the positive dataset rather than very close analogues. We used the values of molecule descriptors and root mean square deviation (RMSD) calculated as the reward function in the reinforcement learning stage in the model. We obtained an initial output of 3,000 structures, which were then automatically filtered to remove molecules bearing structure alerts. The model prioritizes the synthetic feasibility of a small-molecule compound, and the distinctiveness of the compound from other molecules in the literatures and in the patent space. The top 50 candidate moieties generated from the previous step were used as seed structures to search for commercial-available aromatic fragments (similarity ~97%). Lastly, we purchased 10 commercial-available aromatic fragments which were experimentally tested via biolayer interferometry assay (BLI).

## Bio-layer interferometry (BLI) assay

The Octet K2 system (Molecular Device, ForteBIO, USA) is suitable for the characterization of protein-protein or protein-ligand binding kinetics and binding affinity. The assay consists of the following steps: 1. Protein and BLI sensor preparation. The recombinant protein CRM1 (50 μg/mL) was biotinylated in the presence of biotin at room temperature (RT) for 1 hr. Then, the excess biotin was removed through spin desalting columns. The recombinant CRM1 protein was then immobilized on super streptavidin sensors (ForteBIO, USA). The sensors were then blocked, washed, and moved into wells containing various concentrations of the test compounds in kinetic buffer (ForteBIO); 2. BLI experimental process. Automated detection was performed using an Octet K2 (Molecular Devices ForteBIO, USA). Buffer

was added to a 96-well plate, and the plate was transferred to the K2 instrument for analysis by running serially at 25 °C with a shaking speed of 1000 rpm. Baseline readings were obtained in buffer (120 s), associations in wells containing compound (180 s), and dissociation in buffer (180 s). The signals from the following buffer were detected over time. First, each of these concentrations was applied to four sets of experiments: (a) CRM1-immobilized sensor in drug-containing kinetics buffer; (b) Blanked-sensor in drug-containing kinetics buffer; (c) CRM1-immobilized sensor in PBS-kinetics buffer; and (d) Blanked-sensor in PBS-kinetics buffer. The final value was calculated using the equation: (a-c)–(b-d). This method removes interference from the sensor and drug-containing buffers. The binding signals were identified and the results were analyzed using the OctetHT V10.0 software.

## Cell viability assay

Cell viability assays were performed as previously described. Human lymphoma cell lines SNK6 and HANK-1 were seeded into 96-well plates and treated with LFS-1107 in concentrations of 0–800 nM for 72 hr. Cell viability was evaluated using the WST-8-based Cell Counting kit-8 (Beyotime), which was added to the wells and incubated for 3 hr. The absorbance of wells at 450 nm (reference wavelength 610 nm) was measured with a microplate reader (Infinite F50, Tecan).

## Cytotoxicity assay

The Research Ethics Committee (REC) of Dalian University of Technology approved conduct of ex vivo assays with donated human cells (approval number: 2018–023). Normal human peripheral blood mononuclear cells (PBMCs) were obtained from blood samples collected from healthy volunteers. Approval was obtained from The Second Affiliated Hospital of Dalian Medical University institutional review board for these studies. Peripheral blood mononuclear cells were isolated by density gradient centrifugation over Histopaque-1077 (Sigma Diagnostics, St. Louis, MO, USA) at 400 g for 30 min. Isolated mononuclear cells were washed and assayed for total number and viability using Trypan blue exclusion. PBMCs were suspended at $8 \times 10^5$ /mL and incubated in RPMI 1640 medium containing 10% FBS in 24-well plates. Platelets suspended in plasma were collected by apheresis from volunteer donors after obtaining written consent (The Second Affiliated Hospital of Dalian Medical University). After dilution 1:10 in Tyrode's buffer, platelets were incubated for 24 hr at 37 °C with graduated concentrations of test compounds (LFS-1107 or KPT-330) dissolved in DMSO. The $EC_{50}$ values was calculated using GraphPad Prism software.

## Western blot

The cytoplasmic and nucleic protein were extracted separately using the protein extraction kit (Boster) according to the manufacture's introduction. Protein (30–45 µg) was fractionated on a 10–15% acrylamide denaturing gel and transferred onto a PVDF membrane. The membrane was blocked with 5% nonfat dry milk in TBST for 1 hr at room temperature and washed in TBST three times, 5 min each time. The membrane was then incubated with primary antibodies at 1:500 to 1:1000 dilutions overnight at 4 °C. After washing with TBST for 15 min, the membrane was incubated with horseradish peroxidase (HRP) -conjugated secondary antibody at a 1:5000 dilutions for 1 hr at room temperature. After further washing in TBST, the proteins were detected by enhanced chemiluminescence on X-ray film with ECL western blotting detection kit (Thermo Fisher Scientific).

## Nuclear export assays and confocal fluorescence microscopy

293T cells were seeded onto glass bottom cell culture dish at a density of 3000–5000 cells in 1000 µL complete media. After incubation at 37 °C, 5% $CO_2$ for 24 hr, cells were treated with CRM1 inhibitors for 3 hr. Small molecule compounds (KPT-330 and LFS-1107) were serially diluted 1:2 starting from 1 µM in RPMI 1640 medium supplemented with 10% FBS. Following the indicated treatments, cells were fixed for 20 min with 4% paraformaldehyde in PBS. Next, cell membranes were permeabilized by 0.3% Triton X-100 in PBS for 20 min. After blocking with 5% bovine serum albumin (BSA) in PBS for 1 hr at 37 °C, cells were treated with IkB$_\alpha$ antibodies in blocking buffer for 24 hr at 4 °C. Anti-mouse Alexa 594 were used as secondary antibodies. Cell nuclei were stained with DAPI for 20 min. After washing, photomicrographic images were recorded with confocal laser scanning microscope Fluoview FV10i. For cell counts, at least 200 cells exhibiting nuclear, nuclear and cytoplasmic, or cytoplasmic

staining were counted from three separate images. Percentages of N(Nuclear), N/C(Nuclear/Cyto-plasmic) and C(Cytoplasmic) cells were calculated and standard deviations were determined.

## ELISA assay

Cell culture media were collected and centrifuged at for 20 min at 1000 $g$ at 4°Cto remove cell debris. The concentrations of serum cytokines were assayed by the ELISA kit (Elabscience Biotechnology Co., Ltd) according to the manufacturer's instructions. Three experimental replicates were performed for each sample.

## RNA Interference experiment

The siRNA targets were as follows: *XPO*1-siRNA-1 (5'-CCAGCAAAGAAUGGCUCAATT-3'), *XPO1*-siRNA-2(5'-GGAAGAUUCUUCCAAGGAATT-3'), *XPO1*-siRNA-3(5'-CCAGGAGACAGC UAUAUUUTT-3'), and the control target was 5'-UUCUCCGAACGUGUCACGUTT-3', all of which were obtained from Future Biotherapeutics. 293T cells were seeded in 96-well plates at a density of 5000 cells per cell in DMEM medium with 10%FBS the day before transfection. 293T cells were trans-fected with control siRNA or *XPO1*-siRNA according to the manufacturer's instructions. The trans-fected cells were then collected for experiments 48 hr after transfection.

## GSH/GSSG ratio detection

SNK6 cells were seeded onto four plates at a density of 100,000 cells/well in 3 mL complete media. After incubation at 37 °C, 5% CO2 for 24 hr, cells were treated with LFS-1107 for 48 hr. Compound LFS-1107 was serially diluted 1:2 starting from 10 mM in RPMI 1640 medium supplemented with 10% FBS. Following the indicated treatments, cells were tested by GSH and GSSG Assay Kit (Beyotime), which can detect the content of GSH and GSSG in the sample separately. We then calculated the GSH:GSSG ratio of each sample. The results were displayed by GraphPad Prism.

## In vivo efficacy studies

NOD/SCID mice were purchased from the Nanjing Biomedical Research Institute of Nanjing University (NBRI). The mice were randomly divided into three groups (n=6–13 per group, day 0). In brief, SNK6 cells were intravenous injected into SCID mice and the growth of ENKTL cells was monitored every week for two consecutive weeks until the intrasplenic accumulation and proliferation of ENKTL cells. Subsequently, LFS-1107 was intraperitoneal injected to this ENTKL disease model per week (dose: 10 mg/kg). In some experiments, SNK6 cells were pretreated with LFS-1107 (1 μmol/L) or DMSO for 48 hr. Cells were then harvested and resuspended and the cell number was counted using a MuseTM cell analyzer. Animal studies in the present work have been conducted in accordance with the ethical standards and the Declaration of Helsinki. The investigation has been approved by the animal care and use committee of our institution.

## Flow cytometry analysis

Monocytes collected from mouse tissue were stained with FITC anti-human CD45 antibody and PE anti-mouse CD45 antibody. Isotype control antibody was used as a negative control. The Cytomics FC500 flow cytometry (Beckman, USA) was used for analysis.

## Statistical analysis

All continuous variables were compared using one-way ANOVA, followed by Dunnett's test or Tukey's test for multiple comparisons whereas error bars in all figures represent the SEM or SD. * $p<0.05$, **$p<0.01$, ***$p<0.001$, ****$p<0.0001$. Survival curves were compared using the log-rank test with GraphPad software.

## Acknowledgements

Yang's laboratory was supported by the National Natural Science Foundation of China (Grant: 81874301), the Fundamental Research Funds for Central University (Grant: DUT22YG122) and the Key Research project of 'be Recruited and be in Command' in Liaoning Province (Personal Target Discovery for Metabolic Diseases). C Liu's lab was supported by grants from the National Natural

Science Foundation of China (No. U20A20381, 82373063), Shenyang Public Health Science and Technology Programme (No. 22-321-32-18) and Science and Technology Programme of Liaoning Province (No. 2022JH2/20200008). Funding: MOST | National Natural Science Foundation of China (NSFC): He Liu, Meisuo Liu, Xibao Tian, Haina Wang, Jiujiao Gao, Hanrui Li, Zhehuan Zhao, Yu Liu, Xuan Chen, Yongliang Yang, 81874301; MOST | National Natural Science Foundation of China (NSFC): Caigang Liu, U20A20381,82373063 The funders had no role in study design, data collection and interpretation, or the decision to submit the work for publication.

## Additional information

### Competing interests
Caigang Liu: Senior editor, *eLife*. The other authors declare that no competing interests exist.

### Funding

| Funder | Grant reference number | Author |
| --- | --- | --- |
| National Natural Science Foundation of China | 81874301 | He Liu |
| National Natural Science Foundation of China | U20A20381 | Caigang Liu |
| National Natural Science Foundation of China | 82373063 | Caigang Liu |

The funders had no role in study design, data collection and interpretation, or the decision to submit the work for publication.

### Author contributions
He Liu, Formal analysis, Investigation, Methodology; Meisuo Liu, Haina Wang, Jiujiao Gao, Hanrui Li, Zhehuan Zhao, Xuan Chen, Investigation, Methodology; Xibao Tian, Formal analysis, Validation, Investigation, Visualization, Methodology; Yu Liu, Resources, Software, Investigation, Methodology; Caigang Liu, Conceptualization, Resources, Supervision, Methodology, Writing - review and editing; Yongliang Yang, Conceptualization, Resources, Supervision, Writing - original draft, Project administration, Writing - review and editing

### Author ORCIDs
He Liu http://orcid.org/0009-0004-9016-3512
Caigang Liu http://orcid.org/0000-0003-2083-235X
Yongliang Yang http://orcid.org/0000-0003-0449-0599

### Ethics
Human subjects: The Research Ethics Committee (REC) of Dalian University of Technology approved conduct of ex vivo assays with donated human cells (approval number: 2018-023). Normal human peripheral blood mononuclear cells (PBMCs) were obtained from blood samples collected from healthy volunteers. Approval was obtained from The Second Affiliated Hospital of Dalian Medical University institutional review board for these studies.
Animal studies in the present work have been conducted in accordance with the ethical standards and the Declaration of Helsinki. The investigation has been approved by the Research Ethics Committee (REC) of Dalian University of Technology (approval number: 2018-023).

### Decision letter and Author response
Decision letter https://doi.org/10.7554/eLife.80625.sa1
Author response https://doi.org/10.7554/eLife.80625.sa2

# Additional files

## Supplementary files
• MDAR checklist

• Source code 1. Deep reinforcement learning model for molecular de-novo design.

## Data availability
All data generated or analyzed during this study are included in the manuscript and supporting file. Source data files have been provided for Figures 1, 2 and 3 as well as their associated figure supplements.

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
