## [Editor Report]

This study provides important findings on the discovery of a novel CRM1 inhibitor. This study provides compelling data on the identification of a novel CRM1 inhibitor and shows its efficiency against extranodal natural killer/T cell lymphoma cells (ENKTL). The findings from this study will provide motivation on optimizing these novel CRM inhibitors for treatments of ENKTL and will be of broad interest to cancer biology.

---

## [Decision Letter]

**Decision letter after peer review:**

Thank you for submitting your article "Discovery and biological evaluation of a potent small molecule CRM1 inhibitor for its selective ablation of extranodal NK/T cell lymphoma" for consideration by *eLife*. Your article has been reviewed by 2 peer reviewers, and the evaluation has been overseen by a Reviewing Editor and Wafik El-Deiry as the Senior Editor. The reviewers have opted to remain anonymous.

Essential revisions:

1. In general, the image quality of this paper needs to be improved.

2. Needs to provide direct evidence that LFS^-1^107 reversibly inhibits the nuclear export receptor CRM1.

3. What are the cellular activities of the existing inhibitor KPT-330 for ENKTL cell lines, for instance, SNK6 cell line, as compared to the 1107 compound?

*Reviewer #1 (Recommendations for the authors):*

Although the current paper is not ready to be published, the approach is original and I would be more than happy to review a re-submitted paper on the topic.

*Reviewer #2 (Recommendations for the authors):*

This is a rather interesting study that identified a potent small-molecular CRM1 inhibitor for the possible treatment of ENTKL. Below are some specific suggestions which I think may help to improve the manuscript in general,

1. It would be necessary to explain in 1-2 sentences in the main text why the authors choose to use a deep-reinforcement learning model for the design of a CRM1 inhibitor.

2. I am curious about the time cost for the generation of seed structures as well as the whole design process via the deep-reinforcement learning model in the manuscript. Please specify.

3. In lines 74-76, the authors claimed that 50 candidate moieties were used as seed structures to search for commercial-available aromatic fragments. The authors should specify what commercial compound collection was used for the similarity searching.

4. Figure 1—figure supplement 4, is this result for ENKTL lymphoma patients or for general lymphoma patients? Please specify.

5. The authors didn't discuss the results of Figure 2- Supplement 1. The authors should comment in 1-2 sentences in the main text.

6. What are the cellular activities of the existing inhibitor KPT-330 for ENKTL cell lines, for instance, SNK6 cell line, as compared to the 1107 compound?

7. Line 182-183, the authors should provide the reference for the model.

8. To my knowledge, CRM1 may have a spectrum of cargo proteins. Is it possible that 1107 may suppress ENKTL cells via other mechanisms or pathways?

9. The font size for Figure 1 and Figure 2 are too small to read. This should be adjusted and amended.

---

## [Author Response]

Essential revisions:Reviewer #1 (Recommendations for the authors):Although the current paper is not ready to be published, the approach is original and I would be more than happy to review a re-submitted paper on the topic.

This has been fixed. Per the referee’s suggestion, we have move Figure 2 to supplementary data.

Reviewer #2 (Recommendations for the authors):This is a rather interesting study that identified a potent small-molecular CRM1 inhibitor for the possible treatment of ENTKL. Below are some specific suggestions which I think may help to improve the manuscript in general,1. It would be necessary to explain in 1-2 sentences in the main text why the authors choose to use a deep-reinforcement learning model for the design of a CRM1 inhibitor.

This has been fixed. We have elaborated in the main text to explain the motivation of using a deep-reinforcement learning model for the design of LFS^-1^107. Please see below or draft-track-change, line 69-73 for more details.

“Previously, we developed a CRM1 inhibitor LFS-829 via traditional virtual screening and structure-based design approach. Herein, we strive to design a CRM1 inhibitor with strong potency against ENKTL cells through the recently developed AIDD (Artificial Intelligence Drug Discovery) approaches.”

2. I am curious about the time cost for the generation of seed structures as well as the whole design process via the deep-reinforcement learning model in the manuscript. Please specify.

We want to thank the referee for this insightful comment. In fact, the training of the deep reinforcement learning molecule-generation model via ChEMBL library (~2 million compounds) took about 5-6 days on a regular GPU server in our lab. Next, the generation of seed structures (~10,000 structures) took about 3~4 hours. The manual inspection of the seed structures took about 3-4 days. Therefore, in total, it takes about two weeks for the whole design process.

3. In lines 74-76, the authors claimed that 50 candidate moieties were used as seed structures to search for commercial-available aromatic fragments. The authors should specify what commercial compound collection was used for the similarity searching.

Many thanks for this critical comment. We used Σ-Aldrich commercial compound collection (~250,000 compounds) for the similarity searching. Please see draft-track-change, line 79-80 for more details.

4. Figure 1—figure supplement 4, is this result for ENKTL lymphoma patients or for general lymphoma patients? Please specify.

We are grateful for this critical comment. The results of Figure 1—figure supplement 4 were extracted from Oncomine database as gene expression profiling of general lymphoma patients.

5. The authors didn't discuss the results of Figure 2- Supplement 1. The authors should comment in 1-2 sentences in the main text.

Many thanks for this helpful comment. We have elaborated the results of Figure 2- Supplement 1 in the main text. Please see below or draft-track-change, line 140-143 for more details.

“Moreover, proteomics analysis suggests that Biological Process (regulation of cellular component organization) and Molecular Function (protein binding) related to CRM1 were modulated upon the treatment of LFS^-1^107 (Figure 2- Supplement 1).”

6. What are the cellular activities of the existing inhibitor KPT-330 for ENKTL cell lines, for instance, SNK6 cell line, as compared to the 1107 compound?

We want to thank the referee for the kind comment. The cellular activities of KPT-330 for ENKTL cell lines are comparable with LFS^-1^107 (HANK1: IC_50_~36nM; SNK6: IC_50_~26nM;). Specifically, the IC_50_ value of KPT-330 against HANK1 is about ~29nM and the IC_50_ value of KPT-330 against SNK6 is about ~31nM.

7. Line 182-183, the authors should provide the reference for the model.

This has been fixed. The reference has been provided for the model.

8. To my knowledge, CRM1 may have a spectrum of cargo proteins. Is it possible that 1107 may suppress ENKTL cells via other mechanisms or pathways?

We are grateful for this comment. We fully agree with the referee that CRM1 have a spectrum of cargo proteins. Yet, we want to argue that our study captures the most prominent mechanism of actions for LFS^-1^107 towards ENKTL, that is, LFS^-1^107 treatment leads to the nuclear retention of IkBα and consequent strong suppression of NF-κB transcriptional activities. Please see below or draft-track-change, line 162-165 for more details.

“We want to remind the reader that we don’t rule out the possibility that LFS^-1^107 may kill ENKTL cells via other mechanisms due to the broad-spectrum cargo proteins of CRM1. Yet, we want to argue that our study has captured the most prominent mechanism of actions for LFS^-1^107 towards ENKTL.”

9. The font size for Figure 1 and Figure 2 are too small to read. This should be adjusted and amended.

This has been fixed.